# Can Neural Networks Improve Classical Optimization of Inverse Problems?

## Abstract

Finding the values of model parameters from data is an essential task in science. While iterative optimization algorithms like BFGS can find solutions to inverse problems with machine precision for simple problems, their reliance on local information limits their effectiveness for complex problems involving local minima, chaos, or zero-gradient regions. This study explores the potential for overcoming these limitations by jointly optimizing multiple examples. To achieve this, we employ neural networks to reparameterize the solution space and leverage the training procedure as an alternative to classical optimization. This approach is as versatile as traditional optimizers and does not require additional information about the inverse problems, meaning it can be added to existing general-purpose optimization libraries. We evaluate the effectiveness of this approach by comparing it to traditional optimization on various inverse problems involving complex physical systems, such as the incompressible Navier-Stokes equations. Our findings reveal significant improvements in the accuracy of the obtained solutions.

## 1 Introduction

Estimating model parameters by solving inverse problems [Tar05] is a central task in scientific research, from detecting gravitational waves [GH18] to controlling plasma flows [MLA$^+$19] to searching for neutrinoless double-beta decay [AAA$^+$13, AAA$^+$18]. Iterative optimization algorithms, such as limited-memory BFGS [LN89] or Gauss-Newton [GM78], are often employed for solving unconstrained parameter estimation problems [PTVF07]. These algorithms offer advantages such as ease of use, broad applicability, quick convergence, and high accuracy, typically limited only by noise in the observations and floating point precision. However, they face several fundamental problems that are rooted in the fact that these algorithms rely on local information, i.e., objective values $L(x_k)$ and derivatives close to the current solution estimate $x_k$, such as the gradient $\partial L/\partial x|_{x_k}$ and the Hessian matrix $\partial^2 L/\partial x^2|_{x_k}$. Acquiring non-local information can be done in low-dimensional solution spaces, but the curse of dimensionality prevents this approach for high-dimensional problems. These limitations lead to poor performance or failure in various problem settings:

- *Local optima* attract the optimizer in the absence of a counter-acting force. Although using a large step size or adding momentum to the optimizer can help to traverse small local minima, local optimizers are fundamentally unable to avoid this issue.

- *Flat regions* can cause optimizers to become trapped along one or multiple directions. Higher-order solvers can overcome this issue when the Hessian only vanishes proportionally with the gradient, but all local optimizers struggle in zero-gradient regions.

- *Chaotic regions*, characterized by rapidly changing gradients, are extremely hard to optimize. Iterative optimizers typically decrease their step size to compensate, which prevents the optimization from progressing on larger scales.

Submitted to 37th Conference on Neural Information Processing Systems (NeurIPS 2023). Do not distribute.

In many practical cases, a *set* of observations is available, comprising many individual parameter estimation problems, e.g., when repeating experiments multiple times or collecting data over a time frame [CCC+19, DJO+18, GH18, AAA+13, MAL13] and, even in the absence of many recorded samples, synthetic data can be generated to supplement the data set. Given such a set of inverse problems, we pose the question: *Can we find better solutions $x_i$ to general inverse problems by optimizing them jointly instead of individually, without requiring additional information about the problems?*

To answer this question, we employ neural networks to formulate a joint optimization problem. Neural networks as general function approximators are a natural and straightforward way to enable joint optimization of multiple a priori independent examples. They have been extensively used in the field of machine learning [GBCB16], and a large number of network architectures have been developed, from multilayer perceptrons (MLPs) [Hay94] to convolutional networks (CNNs) [KSH12] to transformers [VSP+17]. Overparameterized neural network architectures typically smoothly interpolate the training data [BHM18, BPL21], allowing them to generalize, i.e., make predictions about data the network was not trained on.

It has recently been shown that this generalization capability or *inductive bias* benefits the optimization of individual problems with grid-like solution spaces by implicitly adding a prior to the optimization based on the network architecture [UVL18, HSDG19]. However, these effects have yet to be investigated for general inverse problems or in the context of joint optimization. We propose using the training process of a neural network as a drop-in component for traditional optimizers like BFGS without requiring additional data, configuration, or tuning. Instead of making predictions about new data after training, our objective is to solve only the problems that are part of the training set, i.e., the training itself produces the solutions to the inverse problems, and the network is never used for inference. These solutions can also be combined with an iterative optimizer to improve accuracy. Unlike related machine learning applications [KAT+19, SGGP+20, SFK+21, RT21, SHT22, HKT21, SF18, RPM20, ALGS+22], where a significant goal is accelerating time-intensive computations, we accept a higher computational demand if the resulting solutions are more accurate.

To quantify the gains in accuracy that can be obtained, we compare this approach to classical optimization as well as related techniques on four experiments involving difficult inverse problems: (i) a curve fit with many local minima, (ii) a billiards-inspired rigid body simulation featuring zero-gradient areas, (iii) a chaotic system governed by the Kuramoto–Sivashinsky equation and (iv) an incompressible fluid system that is only partially observable. We compare joint optimization to direct iterative methods and related techniques in each experiment.

## 2 Related work

Neural networks have become popular tools to model physical processes, either completely replacing physics solvers [KAT+19, SGGP+20, SFK+21, RT21] or improving them [TSSP17, UBF+20, KSA+21]. This can improve performance since network evaluations and solvers may be run at lower resolution while maintaining stability and accuracy. Additionally, it automatically yields a differentiable forward process which can then be used to solve inverse problems [SF18, RPM20, ALGS+22], similar to how style transfer optimizes images [GEB16].

Alternatively, neural networks can be used as regularizers to solve inverse problems on sparse tomography data [LSAH20] or employed recurrently for image denoising and super-resolution [PW17]. Recent works have also explored them for predicting solutions to inverse problems [HKT21, SHT22]. In these settings, neural networks are trained offline and then used to infer solutions to new inverse problems, eliminating the iterative optimization process at test time.

Underlying many of these approaches are differentiable simulations required to obtain gradients of the inverse problem. These can be used in iterative optimization or to train neural networks. Many recent software packages have demonstrated this use of differentiable simulations, with general frameworks [HAL+20, SC19, HKT20] and specialized simulators [TLQL21, LLK19].

Physics-informed neural networks [RPK19] encode solutions to optimization problems in the network weights themselves. They model a continuous solution to an ODE or PDE and are trained by formulating a loss function based on the differential equation, and have been explored for a variety

89 of directions [YZWX19, LPY+21, KGZ+21]. However, as these approaches rely on loss terms
90 formulated with neural network derivatives, they do not apply to general inverse problems.

## 3 Reparameterizing inverse problems with neural networks

92 We consider a set of $n$ similar inverse problems where we take *similar* to mean we can express all of
93 them using a function $F(\xi_i \,|\, x_i)$ conditioned on a problem-specific vector $x_i$ with $i = 1, ..., n$. Each
94 inverse problem then consists of finding optimal parameters $\xi_i^*$ such that a desired or observed output
95 $y_i$ is reproduced, i.e.

$$\xi_i^* = \arg\min_{\xi_i} \mathcal{L}(F(\xi_i \,|\, x_i), y_i), \tag{1}$$

96 where $\mathcal{L}$ denotes an error measure, such as the squared $L^2$ norm $|| \cdot ||_2^2$. We assume that $F$ is
97 differentiable and can be approximately simulated, i.e., the observed output $y_i$ may not be reproducible
98 exactly using $F$ due to hidden information or stochasticity.

99 A common approach to finding $\xi_i^*$ is performing a nonlinear optimization, minimizing $\mathcal{L}$ using
100 the gradients $\frac{\partial \mathcal{L}}{\partial F} \frac{\partial F}{\partial \xi_i}$. In strictly convex optimization, many optimizers guarantee convergence to
101 the global optimum in these circumstances. However, when considering more complex problems,
102 generic optimizers often fail to find the global optimum due to local optima, flat regions, or chaotic
103 regions. Trust region methods [Yua00] can be used on low-dimensional problems but scale poorly to
104 higher-dimensional problems. Without further domain-specific knowledge, these methods are limited
105 to individually optimizing all $n$ inverse problems.

106 Instead of improving the optimizer itself, we want to investigate whether better solutions can be found
107 by jointly optimizing all problems. However, without domain-specific knowledge, it is unknown
108 which parameters of $\xi_i$ are shared among multiple problems. We therefore first reparameterize the
109 full solution vectors $\xi_i$ using a set of functions $\hat{\xi}_i$, setting $\xi_i \equiv \hat{\xi}_i(\theta)$ where $\theta$ represents a set of
110 shared parameters. With this change, the original parameters $\xi_i$ become functions of $\theta$, allowing $\theta$
111 to be jointly optimized over all problems. Here, the different $\hat{\xi}_i$ can be considered transformation
112 functions mapping $\theta$ to the actual solutions $\xi_i$, similar to transforming Cartesian to polar coordinates.
113 Second, we sum the errors of all examples to define the overall objective function $L = \sum_{i=1}^{n} \mathcal{L}_i$.

114 For generality, all $\hat{\xi}_i(\theta)$ should be able to approximate arbitrary functions. We implement them as an
115 artificial neural network $\mathcal{N}$ with weights $\theta$: $\hat{\xi}_i(\theta) \equiv \mathcal{N}(x_i, y_i \,|\, \theta)$. Inserting these changes into Eq. 1
116 yields the reparameterized optimization problem

$$\xi_i^* = \hat{\xi}_i(\theta^*), \quad \theta^* = \operatorname{argmin}_\theta \sum_{i=1}^{n} \mathcal{L}(F(\mathcal{N}(x_i, y_i \,|\, \theta) \,|\, x_i), y_i). \tag{2}$$

117 We see that the joint optimization with reparameterization
118 strongly resembles standard formulations of neural network
119 training where $(x_i, y_i)$ is the input to the network and $F \circ$
120 $L$ represents the effective loss function. However, from
121 the viewpoint of optimizing inverse problems, the network
122 is not primarily a function of $(x_i, y_i)$ but rather a set of
123 transformation functions of $\theta$, each corresponding to a fixed
124 and discrete $(x_i, y_i)$. Figure 1 shows the computational graph
125 corresponding to Eq. 2.

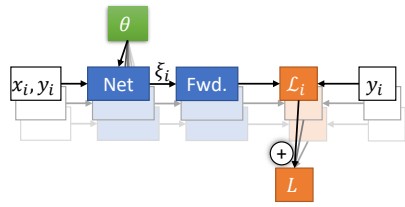

Figure 1: Reparameterized optimization

126 While the tasks of optimizing inverse problems and learning
127 patterns from data may seem unrelated at first, there is a
128 strong connection between the two. The inductive bias of a chosen network architecture, which
129 enables generalization, also affects the update direction of classical optimizers under reparame-
130 terization. This can be seen most clearly if we consider gradient descent steps. There, individual
131 optimization yields the updates $\Delta\xi_i = -\eta \frac{\partial \mathcal{L}_i}{\partial \xi_i}$ with $\eta$ denoting the step size. After reparameterization,
132 the updates are $\Delta\theta = -\eta \sum_i \frac{\partial \mathcal{L}_i}{\partial \xi_i} \frac{\partial \mathcal{N}}{\partial \theta}$. As we can see, $\frac{\partial \mathcal{N}}{\partial \theta}$, which is independent of the specific
133 example, now contributes a large part to the update direction, allowing for cross-talk between the
134 different optimization problems.

Despite the similarities to machine learning, the different use case of this setup leads to differences in the training procedure. For example, while overfitting is usually seen as undesirable in machine learning, we want the solutions to our inverse problems to be as accurate as possible, i.e. we want to "overfit" to the data. Consequently, we do not have to worry about the curvature at $\theta^*$ and will not use mini-batches for training the reparameterization network.

**Supervised training.** Our main goal is obtaining an optimization scheme that works exactly like classical optimizers, only requiring the forward process $F$, $x_i$ in the form of a numerical simulator, and desired outputs $y_i$. However, if we additionally have a prior on the solution space $P(\xi)$, we can generate synthetic training data $\{(x_j, y_i), \xi_j\}$ with $y_j = F(x_j, \xi_j)$ by sampling $\xi_i \sim P(\xi)$. Using this data set, we can alternatively train $\mathcal{N}$ with the supervised objective

$$\tilde{L} = \sum_j ||\mathcal{N}(x_j, y_j) - \xi_j||_2^2. \tag{3}$$

Since $\mathcal{N}$ has the same inputs and outputs, we can use the same network architecture as above and the solutions to the original inverse problems can be obtained as $\xi_i = \mathcal{N}(x_i, y_i)$. While this method requires domain knowledge in the form of the distributions $P(x)$ and $P(\xi)$, it has the distinct advantage of being independent of the characteristics of $F$. For example, if $F$ is chaotic, directly optimizing through $F$ can yield very large and unstable gradients, while the loss landscape of $\tilde{L}$ can still be smooth. However, we cannot expect the inferred solutions to be highly accurate as the network is not trained on the inverse problems we want to solve and, thus, has to interpolate. Additionally, this method is only suited to unimodal problems, i.e. inverse problems with a unique global minimum. On multimodal problems, the network cannot be prevented from learning an interpolation of possible solutions, which may result in poor accuracy.

**Refinement** Obtaining a high accuracy on the inverse problems of interest is generally difficult when the training set size is limited, which can result in suboptimal solutions. This is especially problematic when the global minima are narrow and no direct feedback from $F$ is available, as in the case of supervised training. To ensure that all learned methods have the potential to compete with gradient-based optimizers like BFGS, we pass the solution estimates for $\xi$ to a secondary refinement stage where they are used as an initial guess for BFGS. The refinement uses the true gradients of $F$ to find a nearby minimum of $\mathcal{L}$.

## 4 Experiments

We perform a series of numerical experiments to test the convergence properties of the reparameterized joint optimization. An overview of the experiments is given in Tab. 1 and additional details of all performed experiments can be found in Appendix B. We run each experiment and method multiple times, varying the neural network initializations and data sets to obtain statistically significant results.

To test the capabilities of the algorithms as a black-box extension of generic optimizers, all experiments use off-the-shelf neural network architectures and only require hyperparameter tuning in terms of decreasing the Adam [KB15] learning rate until stable convergence is reached. We then compare the reparameterized optimization to BFGS [LN89], a popular classical solver for unconstrained optimization problems, and to the neural adjoint method, which has been shown to outperform various other neural-network-based approaches for solving inverse problems [RPM20].

**Neural adjoint** The neural adjoint method relies on an approximation of the forward process by a surrogate neural network $S(x_i, \xi_i \,|\, \theta)$. We first train the surrogate on an independent data set

Table 1: Overview of numerical experiments.

| Experiment | $\nabla = 0$ areas | Chaotic | $x_i$ known | $P(\xi)$ known |
|---|---|---|---|---|
| Wave packet localization | No | No | No | Yes |
| Billiards | Yes | No | Yes | No |
| Kuramoto–Sivashinsky | No | Yes | Yes | Yes |
| Incompr. Navier-Stokes | No | Yes | No | Yes |

generated from the same distribution as the inverse problems and contains many examples. We use the same examples as for the supervised approach outlined above but switch the labels to match the network design, $\{(x_i, \xi_i), y_i\}$. After training, the weights $\theta$ are frozen and BFGS is used to optimize $\xi_i$ on the proxy process $\tilde{F}(\xi_i \,|\, x_i) = S(\xi_i, x_i) + B(\xi_i)$ where $B$ denotes a boundary loss term (see Appendix A). With the loss function $\mathcal{L}$ from Eq. 1, this yields the effective objective $\mathcal{L}(F(\xi_i \,|\, x_i), y_i)$ for solving the inverse problems. Like with the other methods, the result of the surrogate optimization is then used as a starting point for the refinement stage described above.

## 4.1 Wave packet localization

First, we consider a 1D curve fit. A noisy signal $u(t)$ containing a wave packet centered at $t_0$ is measured, resulting in the observed data $u(t) = A \cdot \sin(t - t_0) \cdot \exp(-(t - t_0)^2/\sigma^2) + \epsilon(t)$ where $\epsilon(t)$ denotes random noise and $t = 1, ..., 256$. An example waveform is shown in Fig. 2a. For fixed $A$ and $\sigma$, the task is to locate the wave packed, i.e. retrieve $t_0$. This task is difficult for optimization algorithms because the loss landscape (Fig. 2b) contains many local optima that must be traversed. This results in alternating gradient directions when traversing the parameter space, with maximum magnitude near the correct solution.

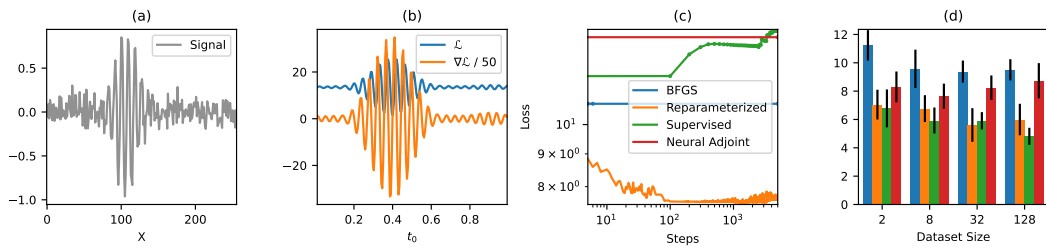

Figure 2: Wave packet localization. **(a)** Example waveform $u(t)$, **(b)** corresponding loss and gradient landscape for $t_0$, **(c)** optimization curves without refinement, **(d)** refined loss $L/n$ by the number of examples $n$, mean and standard deviation over multiple network initializations and data sets.

We generate the inverse problems by sampling random $t_0$ and $\epsilon(t)$ from ground truth prior distributions and simulating the corresponding outputs $u(t) = F\epsilon(t) \,|\, t_0)$. Because the noise distribution $\epsilon(t)$ is not available to any of the optimization methods, a perfect solution with $\mathcal{L} = 0$ is impossible.

Fig. 2c shows the optimization process. Iterative optimizers like BFGS get stuck in local minima quickly on this task. In most examples, BFGS moves a considerable distance in the first iteration and then quickly halts. However, due to the oscillating gradient directions, this initial step is likely to propel the estimate away from the global optimum, leading many solutions to lie further from the actual optimum than the initial guess.

The neural adjoint method finds better solutions than BFGS for about a third of examples for $n = 256$ (see Tab. 2). In many cases, the optimization progresses towards the boundary and gets stuck once the boundary loss $B$ balances the gradients from the surrogate network.

To reparameterize the problem, we create a neural network $\mathcal{N}$ that maps the 256 values of the observed signal $u(t)$ to the unknown value $t_0$. We chose a standard architecture inspired by image classification networks [SZ14] and train it according to Eq. 2. The network consists of five convolutional layers with ReLU activation functions, batch normalization, and max-pooling layers, followed by two fully-connected layers. During the optimization, the estimate of $t_0$ repeatedly moves from minimum to minimum until settling after around 500 iterations. Like BFGS, most examples do not converge to the global optimum and stop at a local minimum instead. However, the cross-talk between different examples, induced by the shared parameters $\theta$ and the summation of the individual loss functions, regularizes the movement in $t_0$ space, preventing solutions from moving far away from the global optimum. Meanwhile, the feedback from the analytic gradients of $F$ ensures that each example finds a locally optimal solution. Overall, this results in around 80% of examples finding a better solution than BFGS.

For supervised training of $\mathcal{N}$, we use the same training data set as for the neural adjoint method. This approach's much smoother loss landscape lets all solution estimates progress close to the ground

truth. However, lacking the gradient feedback from the forward process $\mathcal{F}$, the inferred solutions are slightly off from the actual solution and, since the highest loss values are close to the global optimum, this raises the overall loss during training even though the solutions are approaching the global optima. This phenomenon gets resolved with solution refinement using BFGS.

Fig. 2d shows the results for different numbers of inverse problems and training set sizes $n$. Since BFGS optimizes each example independently, the data set size has no influence on its performance. Variances in the mean final loss indicate that the specific selection of inverse problems may be slightly easier or harder to solve than the average. The neural adjoint method and reparameterized optimization both perform better than BFGS with the reparameterized optimization producing lower loss values. However, both do not scale with $n$ in this example. This feature can only be observed with supervised training whose solution quality noticeably increases with $n$. This is due to the corresponding increase in training set size, which allows the model to improve generalization and does not depend on the number of tested inverse problems. For $n \geq 32$, supervised training in combination with the above-mentioned solution refinement consistently outperforms all other methods.

A detailed description of the network architecture along with additional learning curves, parameter evolution plots as well as the performance on further data set sizes $n$ can be found in Appendix B.1.

## 4.2 Billiards

Next, we consider a rigid-body setup inspired by differentiable billiards simulations of previous work [HAL+20]. The task consists of finding the optimal initial velocity $\vec{v}_0$ of a cue ball so it hits another ball, imparting momentum in a non-elastic collision to make the second ball come to rest at a fixed target location. This setup is portrayed in Fig. 3a and the corresponding loss landscape for a fixed $x$ velocity in Fig. 3b. A collision only occurs if $\vec{v}_0$ is large enough and pointed towards the other ball. Otherwise, the second ball stays motionless, resulting in a constant loss value and $\frac{\partial \mathcal{L}}{\partial \vec{v}_0} = 0$.

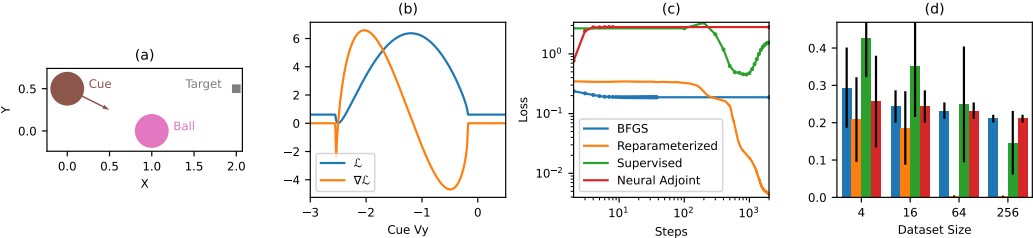

Figure 3: Billiards experiment. **(a)** Task: the cue ball must hit the other ball so that it comes to rest at the target, **(b)** corresponding loss and gradient landscape for $v_y$, **(c)** optimization curves without refinement, **(d)** refined loss $L/n$ by number of examples $n$, mean and standard deviation over multiple network initializations and data sets.

This property prevents classical optimizers from converging if they hit such a region in the solution space. The optimization curves are shown in Fig. 3c. BFGS only converges for those examples where the cue ball already hits the correct side of the other ball.

For reparameterization, we employ a fully-connected neural network $\mathcal{N}$ with three hidden layers using Sigmoid activation functions and positional encoding. The joint optimization with $\mathcal{N}$ drastically improves the solutions. While for $n \leq 32$ only small differences to BFGS can be observed, access to more inverse problems lets gradients from some problems steer the optimization of others that get no useful feedback. This results in almost all problems converging to the solution for $n \geq 64$ (see Fig. 3d).

In this experiment, the distribution of the solutions $P(\vec{v}_0)$ is not available as hitting the target precisely requires a specific velocity $\vec{v}_0$ that is unknown a-priori. We can, however, generate training data with varying $\vec{v}_0$ and observe the final positions of the balls, then train a supervised $\mathcal{N}$ as well as a surrogate network for the neural adjoint method on this data set. However, this is less efficient as most of the examples in the data set do not result in an optimal collision.

The neural adjoint method fails to approach the true solutions and instead gets stuck on the training data boundary in solution space. Likewise, the supervised model cannot accurately extrapolate the true solution distribution from the sub-par training set.

## 4.3 Kuramoto–Sivashinsky equation

The Kuramoto–Sivashinsky (KS) equation, originally developed to model the unstable behavior of flame fronts [Kur78], models a chaotic one-dimensional system, $\dot{u}(t) = -\frac{\partial^2 u}{\partial x^2} - \frac{\partial^4 u}{\partial x^4} - u \cdot \nabla u$. We consider a two-parameter inverse problem involving the forced KS equation with altered advection strength,

$$\dot{u}(t) = \alpha \cdot G(x) - \frac{\partial^2 u}{\partial x^2} - \frac{\partial^4 u}{\partial x^4} - \beta \cdot u \cdot \nabla u,$$

where $G(x)$ is a fixed time-independent forcing term and $\alpha, \beta \in \mathbb{R}$ denote the unknown parameters governing the evolution. Each inverse problem starts from a randomly generated initial state $u(t = 0)$ and is simulated until $t = 25$, by which point the system becomes chaotic but is still smooth enough to allow for gradient-based optimization. We constrain $\alpha \in [-1, 1]$, $\beta \in [\frac{1}{2}, \frac{3}{2}]$ to keep the system numerically stable. Fig. 4a shows example trajectories of this setup and the corresponding gradient landscape of $\frac{\partial \mathcal{L}}{\partial \beta}\|_{\alpha=\alpha^*}$ for the true value of $\alpha$ is shown in Fig. 4b.

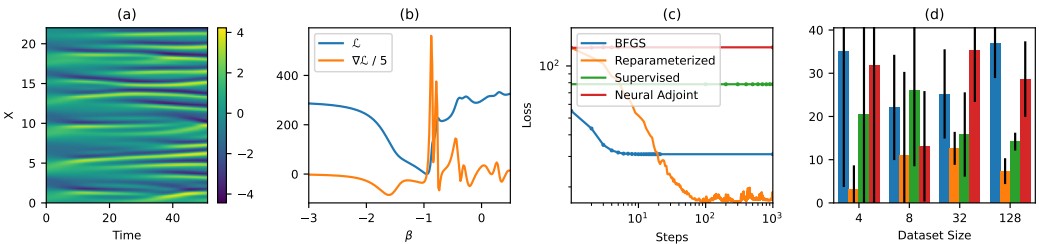

Figure 4: Kuramoto–Sivashinsky experiment. **(a)** Example trajectory, **(b)** corresponding loss and gradient landscape for $\beta$, **(c)** optimization curves without refinement, **(d)** refined loss $L/n$ by number of examples $n$, mean and standard deviation over multiple network initializations and data sets.

Fig. 4c shows the optimization curves for finding $\alpha, \beta$. Despite the complex nature of the loss landscape, BFGS manages to find the correct solution in about 60% of cases. The reparameterized optimization, based on a similar network architecture as for the wavepacket experiment but utilizing 2D convolutions, finds the correct solutions in over 80% of cases but, without refinement, the accuracy stagnates far from machine precision. Refining these solutions with BFGS, as described above, sees the accuracy of these cases decrease to machine precision in 4 to 17 iterations, less than the 12 to 22 that BFGS requires when initialized from the distribution mean $\mathbb{E}[P(\xi)]$.

Supervised training with refinement produces better solutions in 58% of examples, averaged over the shown $n$. The unrefined solutions benefit from larger $n$ on this example because of the large number of possible observed outputs that the KS equation can produce for varying $\alpha, \beta$. At $n = 2$, all unrefined solutions are worse than BFGS while for $n \geq 64$ around 20% of problems find better solutions. With refinement, these number jump to 50% and 62%.

This property also makes it hard for a surrogate network, required by the neural adjoint method, to accurately approximate the KS equation, causing the following adjoint optimization to yield inaccurate results that fail to match BFGS even after refinement.

## 4.4 Incompressible Navier-Stokes

Incompressible Newtonian fluids are described by the Navier-Stokes equations,

$$\dot{u}(\vec{x}, t) = \nu \nabla^2 u - u \cdot \nabla u - \nabla p \quad \text{s.t.} \quad \nabla^2 p = \nabla \cdot v$$

with $\nu \geq 0$. As they can result in highly complex dynamics [BB67], they represent a particularly challenging test case, which is relevant for a variety of real-world problems [Pop00]. We consider

a setup similar to particle imaging velocimetry [Gra97] in which the velocity in the upper half of a two-dimensional domain with obstacles can be observed. The velocity is randomly initialized in the whole domain and a localized force is applied near the bottom of the domain at $t = 0$. The task is to reconstruct the position $x_0$ and initial velocity $\vec{v}_0$ of this force region by observing the initial and final velocity field only in the top half of the domain. The initial velocity in the bottom half is unknown and cannot be recovered, making a perfect fit impossible. Fig. 5a,b show an example initial and final state of the system. The final velocity field is measured at $t = 56$ by which time fast eddies have dissipated significantly.

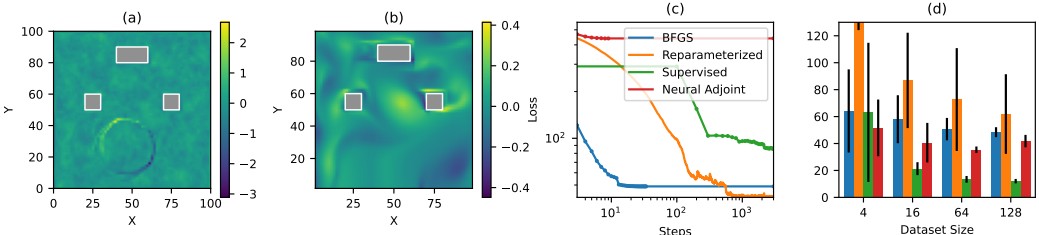

Figure 5: Fluid experiment. **(a,b)** Example initial and final velocity fields, obstacles in gray. Only the upper half, $y \geq 50$, is observed. **(c)** Optimization curves without refinement, **(d)** refined loss $L/n$ by the number of examples $n$, mean and standard deviation over multiple network initializations and data sets.

Fig. 5c shows the optimization curves. On this problem, BFGS converges to some optimum in all cases, usually within 10 iterations, sometimes requiring up to 40 iterations. However, many examples get stuck in local optima.

For joint optimization, we reparameterize the solution space using a network architecture similar to the previous experiment, featuring four 2D convolutional layers and two fully-connected layers. For all tested $n$, the reparameterized optimization produces larger mean loss values than BFGS, especially for small $n$. This results from about 10% of examples seeing higher than average loss values. Nonetheless, 66.7% of the inverse problems are solved more accurately than BFGS on average for $n > 4$.

The neural adjoint method nearly always converges to solutions within the training set parameter space, not relying on the boundary loss. With solution refinement, this results in a mean loss that seems largely independent of $n$ and is slightly lower than the results from direct BFGS optimization. However, most of this improvement comes from the secondary refinement stage which runs BFGS on the true $F$. Without solutions refinement, the neural adjoint method yields inaccurate results, losing to BFGS in 98.2% of cases.

Supervised training does not suffer from examples getting stuck in a local minimum early on. The highest-loss solutions, which contribute the most to $L$, are about an order of magnitude better than the worst BFGS solutions, leading to a much smaller total loss for $n \geq 16$. With solution refinement, 64%, 73% and 72% of examples yield a better solution than BFGS for $n = 16, 64, 128$, respectively.

## 5 Discussion

In our experiments, we have focused on relatively small data sets of between 2 and 256 examples to quantify the worst-case for machine learning methods and observe trends. Using off-the-shelf neural network architectures and optimizers with no tuning to the specific problem, joint optimization finds better solutions than BFGS in an average of 69% of tested problems. However, to achieve the best accuracy, the solution estimates must be passed to a classical optimizer for refinement as training the network to this level of accuracy would take an inordinate amount of time and large data sets. Tuning the architectures to the specific examples could lead to further improvements in performance but would make the approach domain-dependent.

When training data including ground truth solutions are available or can be generated, supervised learning can sidestep many difficulties that complex loss landscapes pose, such as local minima,

Table 2: Fraction of inverse problems for which neural-network-based methods with refinement find better or equal solutions than BFGS. Mean over multiple seeds and all $n$ shown in subfigures (d).

| Experiment | Reparameterized | | Supervised | | Neural Adjoint | |
|---|---|---|---|---|---|---|
| | Better | Equal | Better | Equal | Better | Equal |
| Wave packet fit | **86.0%** | 1.8% | 65.1% | 14.4% | 40.2% | 47.4% |
| Billiards | **61.7%** | 9.0% | 27.0% | 27.2% | 1.6% | 98.4% |
| Kuramoto–Sivashinsky | **62.3%** | 0.0% | 57.7% | 0.0% | 23.9% | 62.2% |
| Incompr. Navier-Stokes | 64.1% | 0.0% | **66.2%** | 0.1% | 56.9% | 0.1% |

alternating gradient directions, or zero-gradient areas. This makes supervised learning another promising alternative to direct optimization, albeit a more involved one.

The neural adjoint method, on the other hand, yields only very minor improvements over BFGS optimization in our experiments, despite the surrogate network successfully learning to reproduce the training data. This is not surprising as the neural adjoint method tries to approximate the original loss landscape which is often difficult to optimize. Improvements over BFGS must therefore come from regularization effects and exposure to a larger part of the solution space. The fact that the neural adjoint method with solution refinement produces similar results almost independent of the number of data points $n$ shows that the joint optimization has little benefit here. Instead, the refinement stage, which treats all examples independently, dominates the final solution quality. Note that the neural adjoint method is purely data-driven and does not require an explicit form for the forward process $F$, making it more widely applicable than the setting considered here.

Tab. 2 summarizes the improvements over classical optimizations for all methods. A corresponding table without solution refinement can be found in Appendix B. Considering that reparameterized optimization is the only network-based method that does not require domain-specific information and nevertheless shows the biggest improvement overall, we believe it is the most attractive variant among the three learned versions. Inverse problems for which reparameterized training does not find good solutions are easy to identify by their outlier loss values. In these cases, one could simply compare the solution to a reference solution obtained via direct optimization, and choose the best result.

**Limitations** We have only considered unconstrained optimization problems in this work, enforcing hard constraints by running bounded parameters through a scaled $\tanh$ function which naturally clamps out-of-bounds values in a differentiable manner.

The improved solutions found by joint optimization come with an increased computational cost compared to direct optimization. The time it took to train the reparameterization networks was 3x to 6x longer for the first three experiments and 22x for the fluids experiment.

# 6 Conclusions and outlook

We have investigated the effects of joint optimization of multiple inverse problems by reparameterizing the solution space using a neural network, showing that joint optimization can often find better solutions than classical optimization techniques. Since our reparameterization approach does not require any more information than classical optimizers, it can be used as a drop-in replacement. This could be achieved by adding a function or option to existing optimization libraries that internally sets up a standard neural network with the required number of inputs and outputs and runs the optimization, hiding details of the training process, network architecture, and hyperparameters from the user while making the gains in optimization accuracy conveniently accessible. To facilitate this, we will make the full source code publicly available.

From accelerating matrix multiplications [FBH$^+$22] to solving systems of linear equations [CHL$^+$23, SSHR19], it is becoming increasingly clear that machine learning methods can be applied to purely numerical problems outside of typical big data settings, and our results show that this also extends to solving nonlinear inverse problems.

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
