# OpenReview forum: "Can Neural Networks Improve Classical Optimization of Inverse Problems?"
_NeurIPS.cc/2023/Conference — Submitted to NeurIPS 2023_

### Official Review · Reviewer_dBXc · 2023-07-06

**Soundness:** 3 good
**Presentation:** 3 good
**Contribution:** 3 good
**Rating:** 5
**Confidence:** 2

**Summary:**

In this paper the authors explore whether better optimization solutions can be found by jointly optimizing several inverse problems together. These inverse problems share a connection as they all can be formulated through a differentiable function $F(\xi_i | x_i)$. The authors implement the joint optimization through a set of NN parameters $\theta$ which connect all the different inverse problem variables $\xi_i = \hat{\xi}_i (\theta)$.

Upon reading the authors' rebuttal to the questions I posed, I am inclined to adjust my score accordingly.

**Strengths:**

* The main problem that the paper is trying to solve is interesting and relevant. As the authors mention in lines 101-105 "generic optimizers often fail to find the global optimum due to local optima, flat regions, or chaotic regions".
* Also the setting expressed in section 3 is not typical which opens up many potential future work leveraging this setting.
* The experimental results are interesting in how they all improve the solutions by increasing $n$ which does provide evidence of cross-talk between the problems.

**Weaknesses:**

* The experiments generate inverse problems synthetically. I would like to see a real-life example of a problem that follows the settings exposed in section 3. I understand that several of the equations have practical applications but in this case I'm referring to a real-life problem that has a naturally occurring (not sampled from a known distribution) set of inverse problems that can be connected through a function $F$.
* The method is not applicable to many problems. I'm not familiar with any ML application that has similar optimization problems that can be pooled together. Put differently, it is unclear to me how restrictive is the setting in section 3 of having a function $F(\xi_i | x_i)$ that expresses all the inverse problems.

**Questions:**

* I understand the BFGS is used as a high-accuracy optimizer. However, in ML SGD is the workhorse as it seems to avoid getting stock in undesirable parts of the solution space. Could you replicate the experiments in section 3 also showing the loss evolution vs steps for SGD?
* What are the runtimes of the experiments that you ran on section 4? I would appreciate a runtime comparison between baselines BFGS and Neural adjoint (not only steps, as the steps could be much more expensive for some methods). I know you make mentions of this on lines 346-348 but a break down for each of the problems and baselines would be ideal. How different are the trained NNs from their initialization?
* Flexible ML models like NNs require several examples to extract the patterns from the data. It is unclear to me how can I learn a NN out of 2 to 256 examples (line 313) as in several problems the neural networks are pretty large too (lines 203-205).
* Could you provide results without the refinement stage (lines 155-161)? I would like to understand how crucial is this last step is to the results obtained. I understand that this stage is done to achieve high accuracy, however it is unclear to me if this stage is actually doing all the work but simply starting from a better initialization coming from the previous stage or if it is actually the opposite and the second stage only improves marginally the results but the first stage gets relatively close to the optimal solution.
* "Supervised learning can sidestep many difficulties that complex loss landscapes pose, such as local minima, alternating gradient directions, or zero-gradient areas" (lines 321-323). Supervised learning is a paradigm of ML where the model's parameters are trained on a dataset with specified targets. As such supervised learning has nothing to do with complex loss landscapes, etc. Could you elaborate on this line and rephrase in the paper?
* How you would you ensure reproducibility of the code as in the README.md it reads: "The code in this directory is preliminary and requires unreleased versions of some libraries" as it is unclear to me what these "unreleased versions" are and whether they would be available as well as your code upon publication (lines 357-358). Could you also add a requirements.txt or setup.py.

**Limitations:**

* The limitations that I see are encapsulated on the questions that I raised above.

---

> ### Author Rebuttal · Authors · 2023-08-08
>
> > I'm not familiar with any ML application that has similar optimization problems that can be pooled together.
>
> We have addressed this point in our general rebuttal above. Our approach can be applied whenever an experiment is performed multiple times or records multiple instances, such as time series data.
>
> > Could you replicate the experiments in section 3 also showing the loss evolution vs steps for SGD?
>
> Thank you for the suggestion! We have now run gradient descent (GD) with adaptive step size as an additional baseline. Example learning curves are shown in Fig. 1 on the attached PDF page. It turns out that GD generally performs worse than BFGS in our experiments. The following table shows the fractions of examples in which GD performs better than or equal to BFGS.
>
> | Experiment             | GD better | Equal to BFGS |
> |------------------------|-----------|-------|
> | Wave packet fit        | 18.4%     | 24.6% |
> | Billiards              | 0.8%      | 30.9% |
> | Kuramoto–Sivashinsky   | 7.0%      | 0.4%  |
> | Incompr. Navier-Stokes | 3.9%      | 0.0%  |
>
> Consequently, our method shows a bigger improvement over GD than BFGS, if slightly.
> In the incompressible fluids experiment with $n=128$, for example, the reparameterized and refined optimization outperforms GD in 88.3% of cases vs 86.0% when compared against BFGS.
>
> > What are the runtimes of the experiments that you ran on section 4?
>
> We have put together a table that includes all training and optimization times. The table can be found in the general rebuttal above. We further discuss the performance comparison to BFGS in our response to reviewer nXp8.
>
> > How different are the trained NNs from their initialization?
>
> This is a very interesting question. We plotted the weight change by layer and data set size in Fig. 2 of the attached PDF page. We can observe that, in all experiments, the weight change grows with increasing data set size, the only real outlier being the final bias of the KS experiment. Partly convolutional networks (wave packet, KS, fluids) concentrate the weight change on the final layers while fully-connected networks (billiards) primarily change the initial layers. We will add these plots along with a discussion to the appendix of our paper.
>
> > It is unclear to me how can I learn a NN out of 2 to 256 examples as in several problems the neural networks are pretty large too.
>
> This question touches on the frontiers of our understanding of ML. Technically, a neural network can be trained with a single example by repeatedly feeding it that data point and backpropagating the residual. Eventually, the network will output the correct result.
> While overparameterization often leads to overfitting and poor generalization in classical optimization, this does not seem to be the case regarding neural networks, see e.g. https://arxiv.org/abs/2105.14368 .
> In our setting, we do not require the networks to generalize well, as there is no separate test set. However, the fact that increasing the data set size improves performance in our experiments shows that the networks generalize to some extent.
>
> > Could you provide results without the refinement stage?
>
> Table 2 in the appendix lists the results without refinement. Also, Figs. 1, 4, 7, 10, 11, and 12 in the appendix show results both with and without the refinement stage.
>
> > it is unclear to me if this stage is actually doing all the work [...] or if it is actually the opposite and the second stage only improves marginally [...].
>
> We have compiled a table listing the fraction of the total loss improvement performed by the network fit. It can be found in our general rebuttal above.
> The table shows that, in all experiments, the first stage (network fit) is responsible for the bulk of the improvement and the refinement stage improves the loss much less overall. We can also see that for larger data set sizes $n$, the network fit contributes even more to the overall improvement while for small data set sizes, the refinement stage is more important.
>
> > I would like to see a real-life example of a problem that follows the settings exposed in section 3.
>
> We agree that this is a very interesting avenue of research. However, we believe that applying this approach to real-life problems would go beyond the scope of this paper. We have plans to pursue this direction in the future.
>
> > (lines 321-323). [...] Could you elaborate on this line and rephrase in the paper?
>
> What we meant here is that the gradients which supervised learning receives are not passed through the simulator and, thus, do not directly depend on the loss landscape that the classical optimizer sees. Supervised learning here simply is linear regression for a given set of training inputs and labels under an $L^2$ loss. With overparameterized networks, this objective almost always results in smooth and stable convergence towards zero training loss. We will reformulate this in the paper.
>
> > How you would you ensure reproducibility of the code as in the README.md it reads [...] Could you also add a requirements.txt or setup.py.
>
> We had to use nightly builds of one library during development, but this has now been resolved with the latest release. We will add a requirements.txt and provide a simple API to run our method on custom inverse problems.
> In the meantime, you should be able to reproduce our experiments with the source code from the supplementary material after installing the following packages:
>
> `torch==2.0.0 tqdm==4.64.1 phiflow==2.4.0 dataclasses==0.6 matplotlib==3.5.1`

---

> > ### Comment · Reviewer_dBXc · 2023-08-18
> >
> > I thank the authors for running some additional experiments which have clarified some questions that I had about the effect of refinement, the actual run times (not only steps) and the evolution of the NN's parameters.
> >
> > I have revisited my score.

---

> > > ### Author Response · Authors · 2023-08-20
> > >
> > > We are grateful for your careful reading of our rebuttal and your revised score! We are glad we could answer your questions and alleviate your concerns.

---

### Official Review · Reviewer_XkkZ · 2023-07-06

**Soundness:** 2 fair
**Presentation:** 3 good
**Contribution:** 2 fair
**Rating:** 4
**Confidence:** 2

**Summary:**

This paper develops a novel approach to gradient-based non-convex optimization. The proposed methodology begins with the reparameterization of the parameter space utilizing neural networks, followed by the application of classical techniques such as BFGS, or alternative Neural Network surrogate models for the forward function, to accomplish the optimization task.

The efficacy of these methods is verified through four distinctive experiments, which include applications to the Kuramoto-Sivashinsky (K-S) equation and the Incompressible Navier-Stokes equation. The results indicate that the reparameterized optimizer delivers enhanced convergence overall.

**Strengths:**

The paper presents a compelling concept of reparametrizing the parameter space of the optimization problem using a neural network and accomplishing optimization via a two-step process that employs the trained/optimized neural network as a preconditioner. However, it remains unclear to the me as to why such a reparameterization is likely to benefit the non-convex optimization procedure. Despite this, the experimental results appear to indicate an enhancement, as evidenced by the four case studies investigated by the author.

**Weaknesses:**

My primary concern regarding this paper pertains to its lack of rigor. The authors do not clearly define the inverse problem that they are attempting to solve, nor do they provide cogent proofs or insights explaining why the introduced reparameterization would aid the optimization process. It is quite plausible that the limited experimental studies offered in this paper lack generalizability, and it's conceivable that there are counterexamples where optimizers, without the incorporation of reparameterization, achieve superior convergence.

**Questions:**

In all four cases, the authors do not explicitly define the inverse problem they're attempting to solve, as well as the specific loss function utilized. This lack of clarity greatly complicates the task of comprehending their methodology.

The paper falls short in normalizing the loss functions - without this normalization, it becomes exceedingly difficult to ascertain whether a loss value, such as "10", carries any physical significance.

The optimization curves presented in the four experiments exhibit some anomalous behaviors. For instance, in Figure 3, the loss of the neural adjoint method escalates with increasing steps, while the supervised method demonstrates oscillation. Could this be attributed to improper tuning of the optimizer's step size?

The paper doesn't provide a direct comparison between the optimized solutions obtained and the ground truth, except for the reported loss function.

**Limitations:**

While the authors recognize that they have not extend the method for constrained optimization problems, I believe there is an inherent limitation in the approach, as it lacks a mechanistic understanding of why such a method would be effective. This comprehension is fundamental for a method to be broadly applicable and reliable.

---

> ### Author Rebuttal · Authors · 2023-08-08
>
> > The authors do not clearly define the inverse problem that they are attempting to solve
>
> Our inverse problems have very simple definitions: in all experiments, the objective is the $L^2$ loss between the target and the simulation output from the solution estimate, see Eq. 2. In the wave packet experiment, the difference is between the estimated and target waveform. In the Billiards experiment, it’s the (x,y) difference between the final position of the ball and the target. In the KS and fluid experiments, it’s the difference in final states. We will add the specific loss functions for each experiment to the experimental details section.
>
> >  [...] nor do they provide cogent proofs or insights explaining why the introduced reparameterization would aid the optimization process
>
> Our method works better than classical optimizers for the same reason neural networks learn to generalize the training data. This “unreasonable effectiveness of neural networks” has been demonstrated empirically many times but our theoretical understanding lags behind. It is likely that future progress made in understanding why neural networks generalize can be directly applied to our results.
>
> > it's conceivable that there are counterexamples where optimizers, without the incorporation of reparameterization, achieve superior convergence.
>
> First, we would like to point out that our experiments cover a broad range of scenarios, and we have never encountered a worsening of results due to reparameterization.
>
> But looking at this from a more theoretical point of view, it also seems unlikely. Remember that classical optimization is also performed as a second stage with our method. To yield worse results, cross-talk between examples would have to move the solution estimates towards more shallow local optima, overcoming the pull along the negative gradient.
> Realistically, this can only occur if the initial guesses were already well-chosen, in which case our method is superfluous anyway. While not impossible, we deem it unlikely that the reparameterized and refined solutions would be worse than those found directly by a classical optimizer in realistic settings.
>
> > The paper falls short in normalizing the loss functions
>
> Our loss functions, namely $L^2$ losses are widely used in machine learning research, as well as science in general. Reporting absolute loss values is standard practice, and it simplifies replication in future work. Furthermore, there is no one correct way to normalize our loss values since they are unbounded and can be zero. The differences in initial loss stem from the initial guess being closer or further from a minimum and do not directly correlate with the difficulty of the specific example.
>
> > in Figure 3, the loss of the neural adjoint method escalates with increasing steps, while the supervised method demonstrates oscillation. Could this be attributed to improper tuning of the optimizer's step size?
>
> We can confirm that neither behavior is caused by an improper step size. You can find the corresponding parameter evolution curves in Fig. 6 of the appendix.
>
> **Neural adjoint**: In all cases, the neural adjoint method converges to a single position in solution space where the optimization reaches machine precision accuracy and then terminates. The actual optimization loss, measured by the surrogate model, decreases during the optimization. However, this is only a proxy for the real loss, which increases.
>
> **Supervised**: While the supervised method does show certain fluctuations in its output, this is because the network is evaluated on a different data set than it is trained on. This is a common phenomenon, related to double descent. Note that the “oscillation” occurs over the course of hundreds to thousands of iterations. A too-large learning rate would result in much higher-frequency oscillations.
>
> > The paper doesn't provide a direct comparison between the optimized solutions obtained and the ground truth, except for the reported loss function.
>
> The distances from ground truth values for all optimization parameters can be found in Figs. 3, 9, and 14 in the appendix where the dashed gray line represents the ground truth solution. These figures cover three of our experiments but we do not have ground truth solutions for the billiards experiment due to how the examples are generated.
>
> Note that the distance in solution space is not a good indicator for the loss, however. In the wave packet experiment, for example, the loss oscillates with increasing distance from the ground truth. By definition, the loss value measures the goodness of solutions and it should be the primary metric for comparing methods.

---

> > ### Comment · Reviewer_XkkZ · 2023-08-21
> >
> > I thank the author for the detailed explaination and revision. I have updated my score.

---

> > > ### Author Response · Authors · 2023-08-21
> > >
> > > Thank you for your careful consideration of our rebuttal! We hope we have resolved your questions and worries.

---

### Official Review · Reviewer_VvA5 · 2023-07-06

**Soundness:** 3 good
**Presentation:** 3 good
**Contribution:** 3 good
**Rating:** 7
**Confidence:** 3

**Summary:**

The manuscript presents a method to reparameterize and solve multiple inverse problems jointly using neural networks. The manuscript tests the proposed method on multiple inverse problems (including some chaotic problems) and compares against Neural Adjoint and BFGS baselines to show measurable performance improvements.

**Strengths:**

* The method is simple, and the authors haven't tuned architecture for problems, which would allow their usage as drop-in replacements.
* Comparison against baselines shows the method provides noticeable improvements.

**Weaknesses:**

* The main downside of these methods is the added training cost (which the authors have mentioned in the limitations section)
    * I would recommend adding the training wall clock times + solving times in a table to give potential users of this method a proper estimate.
* Adding benchmarks for the same problems used in the Neural Adjoint paper would strengthen the paper.

**Questions:**

* The paper uses BFGS to refine the solution. Would it be possible to use the solver as a predictor-corrector and incorporate BFGS in the training pipeline? Similar to methods used in Deep Equilibrium Networks [1] [2].

[1] [Neural Deep Equilibrium Solvers](https://openreview.net/forum?id=B0oHOwT5ENL)

[2] [Continuous Deep Equilibrium Models: Training Neural ODEs Faster by Integrating Them to Infinity](https://arxiv.org/abs/2201.12240)

**Limitations:**

All limitations are clearly stated.

---

> ### Author Rebuttal · Authors · 2023-08-08
>
> > I would recommend adding the training wall clock times + solving times in a table to give potential users of this method a proper estimate.
>
> We have assembled a table that includes all training and optimization times. It is provided in the general rebuttal above.
>
> > Adding benchmarks for the same problems used in the Neural Adjoint paper would strengthen the paper.
>
> That is a good point. However, the Neural Adjoint paper did not use the functional form of the forward process, i.e. the true simulation. When using it, the problems presented there become trivial to solve. As a sanity check, we have now replicated the robotic arm experiment. Fig. 3 in the attached PDF page shows the results. BFGS and gradient descent (GD) manage to reach machine precision accuracy within a couple of iterations while the network approaches take longer to fit the data (3c). The refinement stage then optimizes all examples to machine precision accuracy (3d). While reparameterized fitting successfully solves these experiments, there is no need to use it since they can be solved perfectly with classical optimizers. All inverse problems in our manuscript exhibit non-trivial features, such as local optima, zero-gradient regions, or chaotic behavior.
>
> > The paper uses BFGS to refine the solution. Would it be possible to use the solver as a predictor-corrector and incorporate BFGS in the training pipeline? Similar to methods used in Deep Equilibrium Networks [1] [2].
>
> That’s an interesting idea, but there are two major challenges to overcome. First, the training would be computationally expensive as each network optimization step would require a BFGS solve that itself calls the simulation many times. Second, assuming BFGS converges to a local optimum, the output will always have zero gradient w.r.t. the loss function, so there is no residual to back-propagate. However, as similar approaches have been successfully used in related work, this would be an interesting topic for future work.

---

> > ### Comment · Reviewer_VvA5 · 2023-08-15
> >
> > Thanks for the detailed rebuttal and the updated timings in the rebuttal.
> >
> > > assuming BFGS converges to a local optimum, the output will always have zero gradient w.r.t. the loss function
> >
> > That is a fair point that I missed during my paper review. Thanks for clarifying.
> >
> > I have updated my score.

---

> > > ### Author Response · Authors · 2023-08-16
> > >
> > > Thank you for taking the time to read our rebuttal and for updating your score. We appreciate your constructive feedback and are glad that our points were helpful in addressing your concerns.

---

### Official Review · Reviewer_nXp8 · 2023-07-10

**Soundness:** 3 good
**Presentation:** 4 excellent
**Contribution:** 3 good
**Rating:** 6
**Confidence:** 3

**Summary:**

This paper discusses a novel approach to finding model parameters from data, a crucial task in science. Traditional iterative optimization algorithms like BFGS can accurately solve simple inverse problems, but their reliance on local information can limit their effectiveness in complex situations with local minima, chaos, or zero-gradient regions.

To overcome these issues, the study proposes the idea of jointly optimizing multiple examples. The authors use neural networks to reparameterize the solution space and utilize the training procedure as an alternative to classical optimization. This method is as versatile as traditional optimizers and does not require additional information about the inverse problems, making it compatible with existing general-purpose optimization libraries.

The paper evaluates the effectiveness of this novel approach by comparing it to traditional optimization on a variety of complex inverse problems involving physical systems, such as the incompressible Navier-Stokes equations. The findings show significant improvements in the accuracy of the solutions obtained, suggesting that this method could be a powerful tool for tackling complex inverse problems.

**Strengths:**

1. This paper points out a potential new use case of neural networks and deep learning for optimization instead of existing learning to optimize (L2O) methods, that is to use neural networks as part of classic optimization, trying to learn unknown common structures among problem instances of interest. A major difference is that generalization to unseen instances does not matter.

2. The paper is written in crystal clarity with information in every details about the methods, the experiments and the results. The authors discuss about the results of different methods for each setting. Limitations and outlook to future work are also faithfully discussed.

3. Improvements without refinements look impressive on all settings. Improvements after refinements still look great on the first three settings.

**Weaknesses:**

1. For the 4th setting, Incompressible Navier-Stokes, the proposed reparameterization method gives much higher mean losses than BFGS despite the fact that the majority of problems actually improve over BFGS. Could the authors elaborate more on the potential reasons specific to this experiment setting?

2. Since the mean losses could change with different dataset sizes because of the varying instance difficulty, would it be a better presentation of results with relative error or relative loss? (also a relative improvement could be better for results like Figure 4 in the Appendix.

3. 3~6 times more computational cost for the first three settings and up to 22 times for the fluids could be too high for the benefits achieved. This could be subjective but I hope the authors could provide some discussions or justification.

4. Would the "similarity" requirement be too strict to make the proposed method practically useful? For example, in the wave packet localization setting, the parameters A and $\sigma$ are fixed. Is there a practical application scenario that corresponds to this setting?

**Questions:**

See the weaknesses part.

**Limitations:**

High computation cost and potential lack of practical applicability.

---

> ### Author Rebuttal · Authors · 2023-08-08
>
> > For the 4th setting, [...] reparameterization method gives much higher mean losses than BFGS despite the fact that the majority of problems actually improve over BFGS.
>
> This is an artifact of the L2 loss. Few examples with a high loss can dominate the mean even though most examples have a lower loss. You can see this in Figs. 10 and 11 in the appendix, where we have plotted the distribution of loss values. For the reparameterized optimization, it’s primarily the examples with a large left or right initial velocity $\vec v_0$ that get stuck during the optimization and are resistant to refinement. Despite the 30% higher mean loss, the reparameterized fit finds lower loss solutions in 64% of examples for $n=128$.
>
> > [...] would it be a better presentation of results with relative error or relative loss? (also a relative improvement could be better for results like Figure 4 in the Appendix.
>
> This is a good idea, but it’s hard to realize in practice since it requires a baseline for the difficulty of each problem. The final losses can be zero, so they are not a good reference. Using the initial loss of each problem is possible, but note that the network starts with a slightly different initial guess than the classical optimizer due to the non-zero weight initialization. Nevertheless, we will add the relative loss plots based on the initial guess of BFGS to the appendix but will keep the absolute loss values in the main text because they also facilitate replication of our results.
>
> > 3~6 times more computational cost for the first three settings and up to 22 times for the fluids could be too high for the benefits achieved. This could be subjective but I hope the authors could provide some discussions or justification.
>
> The computational cost is a limitation of our current implementation. However, we would like to point out that we compared our method to a very efficient parallel BFGS implementation that runs the forward process on the GPU. Most users employing a classical solver would likely run a CPU version, looping over the individual examples. For the KS experiment, sequential BFGS solves took 8x longer than the batched solve for $n=16$ and 76x longer for $n=256$. Running this on the CPU increased runtime by an additional 50-60%. Compared to the sequential CPU approach at $n=256$, our method is 18x faster than BFGS.
>
> In any case, our main goal was achieving better results than classical optimizers given access to the same information. We always ran the classical optimizers to convergence, meaning they cannot further improve their solutions given more time. We believe that finding better solutions is worth the extra computation time, especially since BFGS usually is not used in time-critical applications.
>
> Furthermore, we trained the reparameterization networks long enough for the loss to decrease significantly. This is not strictly necessary. For example, training the KS network on $n=256$ for just one-quarter of the training time cited above reduces the fraction of examples with better solutions than BFGS by just 0.3% from 69.1% to 68.8%. However, network fitting (162s) and refinement (129s) together now only take about 60% longer than the pure BFGS optimization (180s).
>
> > Would the "similarity" requirement be too strict to make the proposed method practically useful? For example, in the wave packet localization setting, the parameters A and σ
>  are fixed. Is there a practical application scenario that corresponds to this setting?
>
> We address the applicability of our method in our general comment above.
> As for the wave packet experiment, we chose to fix A and σ to emulate a constrained optimization task with local minima. It is quite common in science applications to fit a function derived from theory with only a handful of free parameters, e.g.
> * fitting voltage and current data to models such as Ohm’s law, Kirchhoff’s laws, etc.
> * fitting temperature and heat exchange data to models such as Fourier’s law, Stefan-Boltzmann law, etc.
> * fitting vibration and noise data to models of damping, resonance, etc.
> * fitting drug concentration and cell response to models of inhibition, activation, etc.
> * fitting trajectories of objects to models of gravity, air resistance, etc.

---

### Author Rebuttal · Authors · 2023-08-08

We thank all reviewers for their valuable feedback and helpful comments!

In answering the reviewer’s questions, we performed additional experiments and created many new figures and tables. The attached PDF page shows gradient descent as an additional baseline for all our experiments (Fig 1), visualizes the network weight changes during fitting (Fig 2), and shows new results for the robotic arm experiment from the neural adjoint paper ([Ren et al., 2020](https://arxiv.org/abs/2009.12919)).

Some reviewers raised concerns about the applicability of our methods since it relies on multiple similar experiments. This was only briefly discussed in the paper, and we’re happy to clarify this point now. Being able to pool experiments is quite common if you consider that most real-world experiments are performed multiple times. E.g., practically all particle physics experiments collect similar data many times. If one detector collects multiple data, they are all influenced by the detector characteristics, and these characteristics could be learned during the optimization. The same is true for many other fields of research, e.g., wind tunnel experiments are typically performed multiple times for a single model, and measurements of different models could also be pooled. Measurements over time, like PIV, can be split into multiple snapshots to be reconstructed jointly.
The setting of solving many similar inverse problems has also been considered in previous ML research, e.g., https://arxiv.org/abs/2205.11912  https://arxiv.org/abs/2206.07681  https://openreview.net/forum?id=HaZuqj0Gvp2 .

Some reviews noted that we had not specified wall-clock training and inference times. We will add the following table which shows all relevant times for the largest tested data set size $n$ in seconds.

| Experiment             | Parallel BFGS | Network fit     | Refinement      | Supervised fit | Supervised Refinement | Surrogate fit   | Neural Adjoint | N.A. Refinement  |
|------------------------|------------------|-----------------|-----------------|----------------|-----------------------|-----------------|----------------|------------------|
| Wave packet fit        | $15.1 \pm 1.0$   | $46.9 \pm 0.2$  | $15.0 \pm 0.2$  | $24.0 \pm 0.2$ | $13.8 \pm 0.7$        | $46.5 \pm 0.2$  | $13.1 \pm 0.5$ | $13.8 \pm 1.7$   |
| Billiards              | $21.5 \pm 1.0$   | $115.2 \pm 0.6$ | $25.3 \pm 0.2$  | $8.8 \pm 0.1$  | $20.9 \pm 1.7$        | $12.9 \pm 0.1$  | $16.5 \pm 2.5$ | $21.8 \pm 1.1$   |
| Kuramoto–Sivashinsky   | $152.8 \pm 11.8$ | $638.8 \pm 3.7$ | $109.3 \pm 7.2$ | $11.4 \pm 0.9$ | $121.5 \pm 64.2$      | $16.4 \pm 0.8$  | $13.7 \pm 2.3$ | $147.6 \pm 12.3$ |
| Incompr. Navier-Stokes | $1858 \pm 95$    | $29510 \pm 637$ | $1270 \pm 205$  | $212 \pm 7$    | $1390 \pm 333$        | $195.6 \pm 0.1$ | $8.7 \pm 1.2$  | $1451 \pm 63$    |

From our manuscript, it was not clear, how much of the improvement in loss is made by the reparameterization network fit and how much the secondary refinement stage contributes.
In the following table, we have compiled the fraction of the total loss decrease achieved by the network fit. The remaining improvement is made by the refinement stage using BFGS. The given fractions are computed per example and then averaged.

| Experiment             | $n=4$                 | $n=8$                 | $n=32$                | $n=128$              |
|------------------------|-----------------------|-----------------------|-----------------------|----------------------|
| Wave packet fit        | $78.5\\% \pm 17.8\\%$ | $89.1\\% \pm 8.8\\%$  | $92.4\\% \pm 3.7\\%$  | $91.7\\% \pm 4.5\\%$ |
| Billiards              | $88.9\\% \pm 13.0\\%$ | $86.8\\% \pm 14.0\\%$ | $92.9\\% \pm 11.2\\%$ | $98.1\\% \pm 2.0\\%$ |
| Kuramoto–Sivashinsky   | $93.4\\% \pm 9.8\\%$  | $96.2\\% \pm 5.9\\%$  | $96.0\\% \pm 2.5\\%$  | $95.9\\% \pm 1.1\\%$ |
| Incompr. Navier-Stokes | $100.0\\% \pm 0.0\\%$ | $99.4\\% \pm 0.5\\%$  | $96.6\\% \pm 3.4\\%$  | $96.8\\% \pm 2.5\\%$ |


We address all other questions and comments raised by the reviews below.

---

### Decision · Program_Chairs · 2023-09-21

**Decision:**

Reject

**Comment:**

Thank you for your submission and active engagement throughout the review period. The reviewers and I are in agreement that the idea of sharing information between inverse problems by reparameterizing them is reasonable, but after careful discussion, we feel that 1) the methodology is largely shared with many existing meta-learning methods that are not properly referenced (see list below), and 2) the experimental results are limited to synthetic data and does not help advance any real-world inverse problem setup (reviewers Xkkz/dBXc). The paper also does not advance the theoretical understanding of these topics and, in the words of reviewer Xkkz, "it is quite plausible that the limited experimental studies offered in this paper lack generalizability, and it's conceivable that there are counterexamples where optimizers, without the incorporation of reparameterization, achieve superior convergence." We recommend for the authors to take these comments into consideration for the next version of the paper.

[1] Khalil, Elias, et al. "Learning combinatorial optimization algorithms over graphs." Advances in neural information processing systems 30 (2017).

[2] Dai, Hanjun, et al. "Neural stochastic dual dynamic programming." arXiv preprint arXiv:2112.00874 (2021).

[3] Yoshua Bengio, Samy Bengio, and Jocelyn Cloutier. Learning a synaptic learning rule. Université de Montréal, Département d’informatique et de recherche opérationnelle, 1990.

[4] Samy Bengio, Yoshua Bengio, Jocelyn Cloutier, and Jan Gecsei. On the optimization of a synaptic learning rule. In Conference on Optimality in Artificial and Biological Neural Networks, 1992.

[5] Thomas Philip Runarsson and Magnus Thor Jonsson. Evolution and design of distributed learning rules. In IEEE Symposium on Combinations of Evolutionary Computation and Neural Networks, 2000.

[6] Marcin Andrychowicz, Misha Denil, Sergio Gomez, Matthew W Hoffman, David Pfau, Tom Schaul, Brendan Shillingford, and Nando De Freitas. Learning to learn by gradient descent by gradient descent. Neural Information Processing Systems (NeurIPS), 2016.

[7] Olga Wichrowska, Niru Maheswaranathan, Matthew W Hoffman, Sergio Gomez Colmenarejo, Misha Denil, Nando Freitas, and Jascha Sohl-Dickstein. Learned optimizers that scale and generalize. In International Conference on Machine Learning (ICML), 2017.

[8] Luke Metz, Niru Maheswaranathan, Jeremy Nixon, Daniel Freeman, and Jascha Sohl-Dickstein. Understanding and correcting pathologies in the training of learned optimizers. In International Conference on Machine Learning (ICML), 2019.

[9] Luke Metz, C. Daniel Freeman, James Harrison, Niru Maheswaranathan, and Jascha SohlDickstein. Practical tradeoffs between memory, compute, and performance in learned optimizers. In Conference on Lifelong Learning Agents (CoLLAs), 2022.

[10] Kaifeng Lv, Shunhua Jiang, and Jian Li. Learning gradient descent: Better generalization and longer horizons. arXiv:1703.03633, 2017.

[11] Yuhuai Wu, Mengye Ren, Renjie Liao, and Roger Grosse. Understanding short-horizon bias in stochastic meta-optimization. In International Conference on Learning Representations (ICLR), 2018.